# Physical Activity Is Associated with Improved Overall Survival among Patients with Metastatic Colorectal Cancer

**DOI:** 10.3390/cancers14041001

**Published:** 2022-02-16

**Authors:** Karel C. Smit, Jeroen W. G. Derksen, Geerard L. O. Beets, Eric J. Th. Belt, Maaike Berbée, Peter Paul L. O. Coene, Hester van Cruijsen, Marjan A. Davidis, Jan Willem T. Dekker, Joyce M. van Dodewaard-de Jong, Annebeth W. Haringhuizen, Helgi H. Helgason, Mathijs P. Hendriks, Ronald Hoekstra, Ignace H. J. T. de Hingh, Jan N. M. IJzermans, Johan J. B. Janssen, Joop L. M. Konsten, Maartje Los, Leonie J. M. Mekenkamp, Peter Nieboer, Koen C. M. J. Peeters, Natascha A. J. B. Peters, Hans J. F. M. Pruijt, Patricia Quarles van Ufford-Mannesse, Ron C. Rietbroek, Anandi H. W. Schiphorst, Arjan Schouten van der Velden, Ruud W. M. Schrauwen, Mark P. S. Sie, Dirkje W. Sommeijer, Dirk J. A. Sonneveld, Hein B. A. C. Stockmann, Marleen Tent, Frederiek Terheggen, Manuel L. R. Tjin-A-Ton, Liselot Valkenburg-van Iersel, Ankie M. T. van der Velden, Wouter J. Vles, Theo van Voorthuizen, Johannes A. Wegdam, Johannes H. W. de Wilt, Miriam Koopman, Anne M. May

**Affiliations:** 1Julius Center for Health Sciences and Primary Care, University Medical Center Utrecht, Utrecht University, 3508 GA Utrecht, The Netherlands; k.c.smit-4@umcutrecht.nl (K.C.S.); h.w.g.derksen-2@umcutrecht.nl (J.W.G.D.); 2Department of Medical Oncology, University Medical Center Utrecht, Utrecht University, 3584 CX Utrecht, The Netherlands; m.koopman-6@umcutrecht.nl; 3Department of Surgery, The Netherlands Cancer Institute, Plesmanlaan 121, 1066 CX Amsterdam, The Netherlands; g.beets@nki.nl; 4GROW School for Oncology and Developmental Biology, Maastricht University, 6200 MD Maastricht, The Netherlands; 5Department of Surgery, Albert Schweitzer Hospital, Albert Schweitzerplaats 25, 3318 AT Dordrecht, The Netherlands; e.j.t.belt@asz.nl; 6Department of Radiotherapy, Maastro Clinic, 6202 NA Maastricht, The Netherlands; maaike.berbee@maastro.nl; 7Department of Surgery, Maasstad Hospital, Maasstadweg 21, 3079 DZ Rotterdam, The Netherlands; coenep@maasstadziekenhuis.nl; 8Department of Medical Oncology, Antonius Hospital, 8600 BA Sneek, The Netherlands; h.vcruijsen@antonius-sneek.nl; 9Department of Medical Oncology, Rivas, Banneweg 57, 4204 AA Gorinchem, The Netherlands; m.davidis@rivas.nl; 10Department of Surgery, Reinier de Graaf Hospital, Reinier de Graafweg 5, 2600 GA Delft, The Netherlands; j.w.t.dekker@rdgg.nl; 11Department of Medical Oncology, Meander Medical Center, Maatweg 3, 3813 TZ Amersfoort, The Netherlands; jm.van.dodewaard@meandermc.nl; 12Department of Medical Oncology, Ziekenhuis Gelderse Vallei, 6710 HN Ede, The Netherlands; haringhuizena@zgv.nl; 13Department of Medical Oncology, Haaglanden Medical Center, 2501 CK Den Haag, The Netherlands; h.helgason@haaglandenmc.nl; 14Department of Medical Oncology, Northwest Clinics, 1800 AM Alkmaar, The Netherlands; m.p.hendriks@nwz.nl; 15Department of Medical Oncology, Ziekenhuisgroep Twente, Zilvermeeuw 1, 7609 PP Hengelo, The Netherlands; r.hoekstra@zgt.nl; 16Department of Surgery, Catharina Hospital, Michelangelolaan 2, 5623 EJ Eindhoven, The Netherlands; ignace.d.hingh@catharinaziekenhuis.nl; 17Department of Surgery, Erasmus MC Cancer Institute, University Medical Center Rotterdam, 3000 AD Rotterdam, The Netherlands; j.ijzermans@erasmusmc.nl; 18Department of Medical Oncology, Canisius Wilhelmina Hospital, 6500 GS Nijmegen, The Netherlands; johan.janssen@cwz.nl; 19Department of Surgery, Viecuri Hospital, Tegelseweg 210, 5912 BL Venlo, The Netherlands; jkonsten@viecuri.nl; 20Department of Medical Oncology, St. Antonius Hospital, 3430 EM Nieuwegein, The Netherlands; m.los@antoniusziekenhuis.nl; 21Department of Medical Oncology, Medisch Spectrum Twente, 7500 KA Enschede, The Netherlands; l.mekenkamp@mst.nl; 22Department of Medical Oncology, Wilhelmina Hospital, 9400 RA Assen, The Netherlands; p.nieboer@wza.nl; 23Department of Surgery, Leiden University Medical Center, University of Leiden, Postzone K6-39 Albinusdreef 2, 2300 RC Leiden, The Netherlands; k.c.m.j.peeters@lumc.nl; 24Department of Medical Oncology, Sint Jans Hospital, Vogelsbleek 5, 6001 BE Weert, The Netherlands; najb.peters@sjgweert.nl; 25Department of Medical Oncology, Jeroen Bosch Hospital, 5200 ME Den Bosch, The Netherlands; h.pruijt@jbz.nl; 26Department of Medical Oncology, Haga Hospital, Els Borst-Eilersplein 275, 2545 AA Den Haag, The Netherlands; p.quarles@hagaziekenhuis.nl; 27Department of Medical Oncology, Rode Kruis Hospital, Vondellaan 13, 1942 LE Beverwijk, The Netherlands; rrietbroek@rkz.nl; 28Department of Surgery, Diakonessenhuis Hospital, Bosboomstraat 1, 3582 KE Utrecht, The Netherlands; aschiphorst@diakhuis.nl; 29Department of Surgery, St. Jansdal Hospital, Wethouder Jansenlaan 90, 3844 DG Harderwijk, The Netherlands; ap.schouten.vander.velden@stjansdal.nl; 30Department of Gastroenterology and Hepatology, Bernhoven Hospital, Nistelrodeseweg 10, 5406 PT Uden, The Netherlands; r.schrauwen@bernhoven.nl; 31Department of Medical Oncology, ZorgSaam Hospital, Wielingenlaan 2, 4535 PA Terneuzen, The Netherlands; m.sie@zzv.nl; 32Department of Medical Oncology, Academisch Medisch Centrum, Meibergdreef 9, 1105 AZ Amsterdam, The Netherlands; d.w.sommeijer@amsterdamumc.nl; 33Department of Medical Oncology, Flevo Hospital, Hospitaalweg 1, 1315 RA Almere, The Netherlands; 34Department of Surgery, Dijklander Hospital, Waterlandlaan 250, 1441 RN Purmerend, The Netherlands; d.j.a.sonneveld@westfriesgasthuis.nl; 35Department of Surgery, Spaarne Hospital, 2000 AK Haarlem, The Netherlands; stockmann@spaarnegasthuis.nl; 36Department of Medical Oncology, Treant Hospital, 7800 RA Emmen, The Netherlands; m.tent@treant.nl; 37Department of Medical Oncology, Bravis Hospital, Boerhaavelaan 25, 4708 AE Roosendaal, The Netherlands; f.terheggen@bravis.nl; 38Department of Medical Oncology, Rivierenland Hospital, 4000 HA Tiel, The Netherlands; tjin@zrt.nl; 39Department of Medical Oncology, Maastricht University Medical Center, 6202 NA Maastricht, The Netherlands; liselot.van.iersel@mumc.nl; 40Department of Medical Oncology, Tergooi Hospital, Van Riebeeckweg 212, 1213 XZ Hilversum, The Netherlands; avandervelden@tergooi.nl; 41Department of Surgery, Ikazia Hospital, Montessoriweg 1, 3083 AN Rotterdam, The Netherlands; wj.vles@ikazia.nl; 42Department of Medical Oncology, Rijnstate Hospital, Postus 9555, 6800 TA Arnhem, The Netherlands; tvanvoorthuizen@rijnstate.nl; 43Department of Surgery, Elkerliek Hospital, Wesselmanlaan 25, 5707 HA Helmond, The Netherlands; jwegdam@elkerliek.nl; 44Department of Surgery, Radboud University Medical Center, University of Nijmegen, 6500 HB Nijmegen, The Netherlands; hans.dewilt@radboudumc.nl

**Keywords:** metastatic colorectal cancer, physical activity, all-cause mortality, survival

## Abstract

**Simple Summary:**

Physical activity is linked to longer survival among non-metastasized colorectal cancer patients. It is unclear if physical activity is also beneficial for survival among patients with metastatic colorectal cancer. We researched this question in our study of 293 patients with metastatic colorectal cancer. We found that participants who reported higher levels of physical activity at diagnosis lived longer compared to patients who reported low activity levels. Furthermore, adherence to the physical activity guidelines for cancer survivors was related to prolonged survival. Our findings suggest that patients with metastatic colorectal cancer also benefit from being physically active. Future studies are needed to investigate whether improving exercise levels after diagnosis of metastasis is also beneficial and what kind of exercise interventions are most optimal for possibly improving survival time of patients with metastatic colorectal cancer.

**Abstract:**

Regular physical activity (PA) is associated with improved overall survival (OS) in stage I–III colorectal cancer (CRC) patients. This association is less defined in patients with metastatic CRC (mCRC). We therefore conducted a study in mCRC patients participating in the Prospective Dutch Colorectal Cancer cohort. PA was assessed with the validated SQUASH questionnaire, filled-in within a maximum of 60 days after diagnosis of mCRC. PA was quantified by calculating Metabolic Equivalent Task (MET) hours per week. American College of Sports and Medicine (ACSM) PA guideline adherence, tertiles of moderate to vigorous PA (MVPA), and sport and leisure time MVPA (MVPA-SL) were assessed as well. Vital status was obtained from the municipal population registry. Cox proportional-hazards models were used to study the association between PA determinants and all-cause mortality adjusted for prognostic patient and treatment-related factors. In total, 293 mCRC patients (mean age 62.9 ± 10.6 years, 67% male) were included in the analysis. Compared to low levels, moderate and high levels of MET-hours were significantly associated with longer OS (fully adjusted hazard ratios: 0.491, (95% CI 0.299–0.807, *p* value = 0.005) and 0.485 (95% CI 0.303–0.778, *p* value = 0.003), respectively), as were high levels of MVPA (0.476 (95% CI 0.278–0.816, *p* value = 0.007)) and MVPA-SL (0.389 (95% CI 0.224–0.677, *p* value < 0.001)), and adherence to ACSM PA guidelines compared to non-adherence (0.629 (95% CI 0.412–0.961, *p* value = 0.032)). The present study provides evidence that higher PA levels at diagnosis of mCRC are associated with longer OS.

## 1. Introduction

A physically active lifestyle is broadly considered to be related to a decreased risk of developing colorectal cancer (CRC), with evidence being especially strong for colon cancer [1,2,3,4,5]. After a cancer diagnosis, several studies show beneficial associations between high levels of physical activity (PA) and reduced mortality in stage I–III CRC patients as well [6,7,8,9,10,11].

Evidence for an association between PA and survival among metastatic colorectal cancer (mCRC) patients, however, is sparse, and the described associations are less uniform. Three studies provided secondary subgroup analyses, including small numbers of mCRC patients with conflicting results [12,13,14].

One large prospective study of patients with mCRC, embedded in a randomized phase III trial, reported a longer progression-free survival (PFS) and lower risk of treatment-related toxicities with higher total physical activity levels. More non-vigorous activity was associated with longer PFS and overall survival (OS), vigorous activity and walking were not [15]. However, as also noted by the authors, selective enrollment in a trial context may affect the generalizability of these results, creating uncertainty as to what extent the results can be extrapolated to the general patient population.

Expanding current knowledge about the association between PA and OS is relevant, as it may help patients to make informed choices regarding their (change of) lifestyle, which is an important question facing a significant proportion of patients [16]. Furthermore, if there is indeed an association between PA and OS in patients with mCRC, this could inform future exercise intervention studies investigating the potential to prolong survival. Previous studies already showed feasibility and safety of exercise interventions in advanced cancer patients in general [17] and the mCRC population specifically [18].

In this study, we aim to provide further evidence of the association between prediagnostic PA and OS among patients with mCRC who participate in a large, prospective cohort study.

## 2. Materials and Methods

### 2.1. Study Sample

Data were obtained from patients participating in the Prospective Dutch Colorectal Cancer (PLCRC) cohort. PLCRC is a prospective multidisciplinary nationwide observational cohort in the Netherlands for which all patients over eighteen years of age with a histologically proven or strong clinical suspicion of CRC are eligible for inclusion. Currently, sixty-one of the sixty-nine Dutch hospitals participate in this initiative. As an informed consent option, patients are asked to consent to receiving repeated questionnaires on health-related topics. A detailed description of the cohort design is published elsewhere [19]. For the current analysis, all patients who completed a questionnaire within sixty days of diagnosis of first metastasis were included. Clinical data on tumor characteristics and treatment information were obtained from the Netherlands Cancer Registry (NCR). Date of first metastasis was defined as date of histological confirmation or date of first imaging of metastasis if no histological proof was obtained. Standardized differences were calculated to quantify the magnitude of differences in patient characteristics between our study population and the general Dutch population of mCRC patients, and between our study population and all mCRC PLCRC participants (i.e., including patients that did not consent to filling out questionnaires). Values greater than 0.20 indicate a large imbalance, while values between 0.10 and 0.20 indicate a small imbalance, and standardized differences less than 0.10 indicate a negligible imbalance [20].

### 2.2. Assessment of Mortality

The primary endpoint of this study is overall survival (OS). Vital status was obtained from an annual data linkage with the municipal population registry on 1 February 2021. Overall survival was defined as the interval from diagnosis of metastatic disease to all-cause death, or the date of the last follow-up was used for censoring (1 February 2021).

### 2.3. Assessment of Physical Activity

PA information was obtained using the validated Dutch short questionnaire to assess health-enhancing physical activity (SQUASH) [21]. The general purpose of the SQUASH is to assess the amount and intensity of habitual PA during an average week in the past months. Questions are divided in four activity domains: commuting activities, sports and leisure time activities, household activities, and activities at work and school.

The total amount of time spent weekly on each activity and each activity domain was calculated. All activities were given a Metabolic Equivalent of Task (MET) score, based on the updated Ainsworth compendium of physical activities [22], and categorized as light-intensity (<3.0 METs), moderate-intensity (3.0–5.9 METs), and vigorous-intensity (≥6.0 METs).

MET-hours were calculated by multiplying the time spent on activities with their assigned MET values. Weekly time spent on moderate-to-vigorous-intensity physical activity (MVPA) was calculated, which contained all activities with ≥3.0 METs. A subset was also made, containing only sport and leisure time MVPA (MVPA-SL). Lastly, Dutch PA guideline adherence was assessed, which correspond to the American College of Sports and Medicine (ACSM) guidelines for cancer survivors, i.e., resistance training at least twice a week, and a minimum of 150 min aerobic exercise per week [23,24].

### 2.4. Assessment of other Study Parameters

Age, sex, primary tumor location, metastatic sites, synchronicity of metastasis, surgery of primary tumor, metastasectomy and systemic treatment information were obtained from the NCR. Synchronous metastasis was defined as having stage IV cancer at first diagnosis. Exposure to systemic treatment types was defined as having received at least one dose of a treatment modality. Total days on systemic treatment was calculated as the cumulative sum of intervals receiving systemic therapy. If no end date was registered, date of censoring was used as end date. Body mass index (BMI) was calculated using the standard kg/m^2^ calculation, based on self-reported height and weight within sixty days of diagnosis of first metastasis.

### 2.5. Statistical Analyses

Cox proportional hazards models were used to analyze associations between PA and OS. PA was assessed using continuous data and tertiles for weekly MET-hours, MVPA, and MVPA-SL (henceforth: tertile 1 = low; tertile 2 = moderate; and tertile 3 = high level of PA), with low being the reference category. Guideline adherence was assessed as a categorical variable (yes, no), with ‘no’ being the reference category. The proportional hazards assumption was tested for PA variables, both visually and by using scaled Schoenfeld residuals.

Two multivariable cox models were applied. The first model (henceforth: adjusted model) was adjusted for characteristics at diagnosis: age (continuous), sex (female, male), BMI (continuous), primary tumor location (left, right, rectum, other), metastatic sites (1, >1), liver-only metastasis (yes or no) and metastasis pattern (synchronous vs. metachronous). The second model (henceforth: fully adjusted model) was adjusted for the same variables, and contained additional adjustment for treatment-related factors: surgery of primary tumor (yes—before diagnosis of 1st metastasis, yes—after diagnosis of 1st metastasis, or no) and metastasectomy (yes or no). Sensitivity analyses were performed to reduce the probability of reverse causation by repeating our Cox models without patients who died within six months after diagnosis of 1st metastasis. All data were analyzed using SPSS version 26 [25] and R version 4.0.3. [26]. Survival analyses were performed with the R survival package, version 3.2–13 [27] and survival plots were created with the R survminer package, version 0.4.9 [28]. All statistical tests were two-sided with an alpha level of 0.050.

## 3. Results

### 3.1. Patient Characteristics

In total, 306 patients completed a SQUASH questionnaire within sixty days of diagnosis of first metastasis. Thirteen participants were excluded from further analysis due to incomplete questionnaires, i.e., more than two of the four domains were missing, leading to a final study population of *n* = 293.

Table 1 shows baseline and treatment characteristics by tertiles of weekly MET-hours, and for the total study population. The majority of patients were male (67.2%) and mean age was 62.9 years (standard deviation (SD) 10.6). Table A1 shows standardized differences between our study population and both the general Dutch mCRC population and the entire PLCRC mCRC population. Compared to the general population, our study population showed a large difference in age (63 vs. 68 years), sex (67% vs. 57% male), and primary tumor localization (39% vs. 28% rectum). Compared to the entire PLCRC mCRC population, our study population showed a small difference in sex (67% vs. 61% male) and negligible differences in age (63 vs. 62 years) and primary tumor localization (39% vs. 39% rectum) [29]. When comparing tertiles, patients with high PA levels (tertile 3) were younger, less likely to have right-sided tumors, and were more likely to have synchronous metastases and to receive a metastasectomy. No clear differences in systemic treatment exposure were seen, but the median total days on treatment was evidently lower in patients with low PA levels (tertile 1, median of 169 days vs. a median of 223 and 215 in tertiles 2 and 3).

### 3.2. Associations of Physical Activity with OS

Figure 1 shows Kaplan–Meier curves of OS by tertiles of MET-hours, MVPA and MVPA-SL, and by ACSM PA guideline adherence. A total of 106 deaths occurred during a median follow-up time of 18.8 months. Based on the Schoenfeld test, proportionality assumption was met for MET-hours (*p* value = 0.202), MVPA (*p* value = 0.164), MVPA-SL (*p* value = 0.294), and guideline adherence (*p* value = 0.257).

Table 2 shows Hazard Ratios (HRs) of the four assessed PA determinants for the univariate and for both multivariable adjusted Cox proportional hazard models.

#### 3.2.1. Metabolic Equivalent Task Hours

An increase in weekly MET-hours was consistently significantly associated with improved survival across all models (univariate HR 0.994 (95% CI 0.990–0.997, *p* value < 0.001); adjusted HR 0.995 (95% CI 0.991–0.998, *p* value = 0.001); and fully adjusted HR 0.995 (95% CI 0.991–0.998, *p* value < 0.001)). Median (IQR) MET-hours per week was 33.1 (10.0, 47.4) for the group with low levels, 89.9 (79.4, 104) for the group with moderate levels, and 166 (140, 199) for the group with high levels. Both moderate and high MET-hour levels were consistently significantly associated with improved survival compared to low levels across all models. Compared to low levels of MET-hour/week, the fully adjusted HR for moderate levels of MET-hours per week was 0.491 (95% CI 0.299–0.807, *p* value = 0.005), and the adjusted HR was 0.448 (95% CI 0.271–0.741, *p* value = 0.002). For high levels of MET-hours per week compared to low levels, the fully adjusted HR was 0.485 (95% CI 0.303–0.778, *p* value < 0.001), and the adjusted HR was 0.491 (95% CI 0.306–0.790, *p* value <0.001).

#### 3.2.2. Moderate and Vigorous Physical Activity

An increase in weekly hours spent on MVPA was consistently significantly associated with improved survival across all models (univariate HR 0.976 (95% CI 0.958–0.994, *p* value = 0.010); adjusted HR 0.975 (95% CI 0.957–0.994, *p* value = 0.010); fully adjusted HR 0.973 (95% CI 0.955–0.992, *p* value = 0.006)). Median (IQR) hours per week spent on MVPA was 0.5 (0.0, 2.4) for the group with low levels, 7.5 (5.1, 10.3) for the group with moderate levels, and 21.6 (16.1, 31.4) for the group with high levels. High levels of MVPA were significantly associated with longer survival compared to low levels (fully adjusted HR 0.476 (95% CI 0.278–0.816, *p* value = 0.007) and adjusted HR 0.491 (95% CI 0.288–0.836, *p* value = 0.088)). Moderate levels of MVPA were not significantly associated with longer OS compared to low levels (fully adjusted HR 0.889 (95% CI 0.556–1.423, *p* value = 0.625), adjusted HR 0.916 (95% CI 0.575–1.459, *p* value = 0.711)).

#### 3.2.3. Sport and Leisure Time Moderate and Vigorous Physical Activity

An increase in weekly hours spent on MVPA-SL was consistently significantly associated with improved survival across all models (univariate HR 0.965 (95% CI 0.938–0.993, *p* value = 0.015); adjusted HR 0.955 (95% CI 0.926–0.986, *p* value = 0.004); and fully adjusted HR 0.957 (95% CI 0.927–0.988, *p* value = 0.007)). Median (IQR) hours per week spent on MVPA-SL was 0.0 (0.0, 0.3) for the group with low levels, 4.2 (2.5, 5.5) in the group with moderate levels, and 14.0 (10.5, 18.5) in the group with high levels. High levels of MVPA-SL were significantly associated with improved survival compared to low levels (fully adjusted HR 0.389 (95% CI 0.224–0.677, *p* value < 0.001) and adjusted HR 0.384 (95% CI 0.223–0.661, *p* value < 0.001)). Moderate levels of MVPA-SL were not associated with a significant difference in OS compared to low levels (fully adjusted HR 0.737 (95% CI 0.462–1.175, *p* value = 0.200) and adjusted HR 0.769 (95% CI 0.480–1.230, *p* value = 0.273)).

#### 3.2.4. ACSM Physical Activity Guideline Adherence

Thirty-eight percent (112/293) of participants adhered to ACSM PA guidelines. Guideline adherence was significantly associated with improved survival in the univariate model (HR 0.628, 95% CI = 0.417-0.945, *p* value = 0.026) and in the fully adjusted model (HR 0.629, 95% CI = 0.412–0.961, *p* value = 0.032), but not in the adjusted model (HR 0.666, 95% CI = 0.439–1.009, *p* value = 0.055).

#### 3.2.5. Sensitivity Analyses

Table A2 shows HRs of the four assessed PA determinants with exclusion of participants that died within six months (*n* = 8). This yielded comparable results for all models with MET-hours, MVPA, and MVPA-SL as determinants. The adjusted model with ACSM PA guideline adherence showed a significant association 0.628 (95% CI 0.407–0.969, *p* value = 0.036), compared to a non-significant association in the primary analysis.

## 4. Discussion

In this prospective observational cohort of mCRC patients, we found that higher weekly total PA (MET-hours), MVPA, and MVPA-SL at diagnosis of first metastasis was significantly associated with prolonged survival time compared to low levels. When comparing tertiles, significant associations were seen for high levels of MVPA and MVPA-SL, and high and moderate levels of total PA (MET-hours per week). Significant associations with increased survival time were also seen for ACSM PA guideline adherence compared to non-adherence. Greatest risk reductions were seen for high levels of MVPA-SL compared to low levels.

To date, the evidence for the relationship between PA and survival in mCRC patients is limited. A previous study by Guercio et al. showed an association between greater non-vigorous activity and improved survival in this patient group, but not of total physical activity [15]. Several other studies have reported data from subgroup analyses of patients with stage IV CRC. One study showed that walking was related to longer OS in mCRC patients [12], whereas two other studies found no associations with survival [13,14]. However, the latter two studies were limited by low numbers. In our study including almost 300 patients with mCRC, we observed clear associations between multiple types of PA and survival.

Furthermore, we investigated PA by means of tertiles, instead of dichotomizing PA levels as in the majority of studies. It could be the case that the cut-off for the most active group in other studies simply was not high enough. This is illustrated by the fact that moderate PA levels show little to no difference in survival time compared to low levels of PA for both MVPA and MVPA-SL in our analyses. In this light however, it is important to note that comparing PA associations of different studies poses challenges due to heterogeneity in assessment of physical activity [21,30,31]. These differences are reflected in reported median weekly MET-hours ranging from 10 to 100/week for different questionnaires [8,32,33]. Besides these differences in assessment, interpretation of results for specific activities/activity types poses the additional challenge that distribution of activity types is known to be different per country [34].

Several underlying biological mechanisms might be involved in beneficial associations between PA and survival, including hyperinsulinemia, inflammation, and obesity [35,36]. Reduction in treatment-related toxicities is also described in physically active patients, which could contribute to increased survival time [37]. This might also have occurred in our study, based on the differences in median total time on treatment between the group with low levels of MET-hours, and the groups with moderate and high levels of MET-hours. Future analyses using repeated measures of PA and more detailed analysis of (changes in) systemic treatment will provide additional and valuable insight into associations with survival time when maintaining or increasing PA levels after mCRC diagnosis. These analyses will also inform future randomized controlled trials investigating the effects of physical activity on mCRC outcomes.

Strengths of our study include the large nation-wide design and population-based study sample, reflecting a more heterogeneous mCRC population as compared to trial-based study samples. The design of the PLCRC cohort made it possible to compare our study sample to all mCRC participants, showing no notable differences in age or tumor characteristics. Detailed and highly complete demographic and clinical baseline variables allowed us to adjust for known prognostic factors. Information on systemic therapy made it possible to compare exposure to treatment modalities, showing comparable percentages for MET-hour tertiles, whereas data on surgical treatment (including metastasectomies) allowed us to account for beneficial effects on survival during follow-up. However, the association between PA and longer survival was virtually unchanged after adjustment for cancer treatment.

This study has some notable limitations to consider as well. First, as this is an observational study with a single measure of PA, the association between PA and OS is at risk for reverse causation. We cannot rule out that a low level of PA is an indicator of worse disease, although we adjusted the analyses for prognostic factors. Additionally, analyses with exclusion of participants who died within six months did not alter our results. Still, residual confounding cannot be ruled out. Furthermore, although mean age from our study sample is higher compared to phase-III clinical trials in mCRC, participants are, on average, 5 years younger than the general mCRC population [29,38], thereby possibly still limiting generalizability of our results. Additionally, our follow-up time was limited with a median follow-up time of 18.8 months. In regard to assessment of our determinant of interest, it should be noted that self-report of PA is inherently vulnerable to misclassification. Subsequently, although the SQUASH questionnaire asks respondents to think about a normal week in the past few months, reporting of baseline activity within sixty days of diagnosis of first metastasis could have been influenced by variance in cancer disease presentation. Nevertheless, the SQUASH questionnaire is a widely used and validated tool to measure habitual PA [21].

## 5. Conclusions

Higher levels of physical activity at mCRC diagnosis are significantly associated with improved survival. Associations were most pronounced with high levels of moderate to vigorous sport and leisure time physical activity. Future observational studies with repeated PA assessments and randomized studies should assess whether increasing PA levels after mCRC diagnosis reduces mortality risk, and investigate optimal exercise interventions and guidance for mCRC patients.

## Figures and Tables

**Figure 1 cancers-14-01001-f001:**
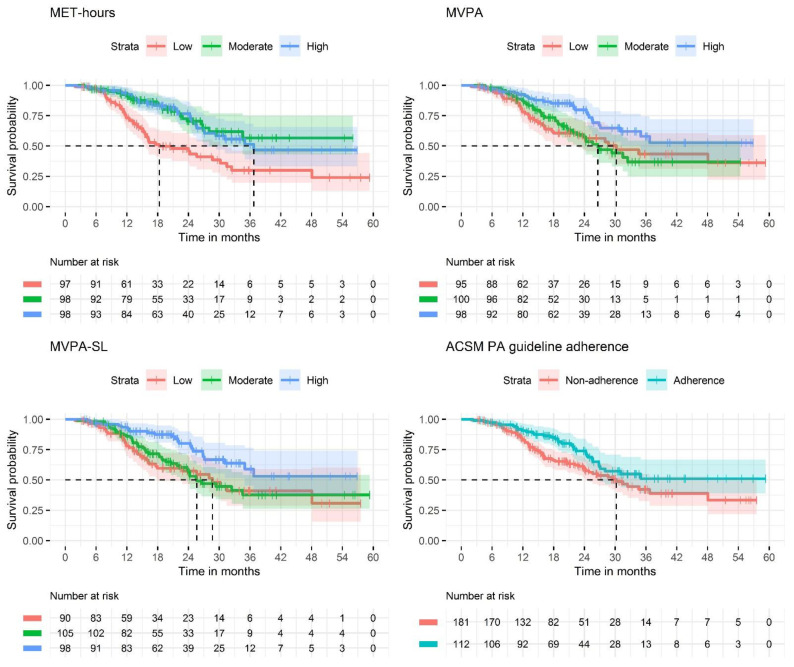
Kaplan–Meier survival curves, stratified by the following physical activity determinants: tertiles of Metabolic equivalent of task (MET) hours; tertiles of moderate and vigorous physical activity (MVPA); tertiles of sport and leisure time moderate and vigorous physical activity (MVPA-SL); and American College of Sports and Medicine (ACSM) physical activity (PA) guideline adherence. Dotted lines represent median survival time when reached.

**Table 1 cancers-14-01001-t001:** Patient characteristics by tertiles of MET-hours per week, and for the total study sample.

Characteristics	Low PA (*n* = 97)	Moderate PA (*n* = 98)	High PA (*n* = 98)	Total (*n* = 293)
Follow-up time in months, median (IQR)	14.7 (10.6, 22.9)	19.9 (13.0, 26.6)	21.3 (15.8, 30.0)	18.8 (12.4, 26.6)
MET-hours/week, median (p2.5, p97.5)	33.1 (0, 60.9)	89.9 (63.8, 119)	166 (122, 303)	90.0 (0, 281)
MVPA hours/week, median (p2.5, p97.5)	1.50 (0, 13.7)	6.8 (0, 25.4)	19.9 (5.0, 60.6)	7.5 (0, 47.4)
MVPA-SL hours/week, median (p2.5, p97.5)	0 (0, 12.6)	4.21 (0, 22.6)	11.1 (1.0, 43.7)	4.5 (0, 31.7)
Adheres to ACSM PA guideline, *n* (%)	4 (4.1%)	43 (43.9%)	65 (66.3%)	112 (38.2%)
Sex, *n* (%)				
Male	67 (69.1%)	64 (65.3%)	66 (67.3%)	197 (67.2%)
Female	30 (30.9%)	34 (34.7%)	32 (32.7%)	96 (32.8%)
Age in years, mean (SD)	66.4 (10.4)	62.1 (10.1)	60.1 (10.3)	62.9 (10.6)
BMI, mean (SD)	25.5 (4.6)	25.2 (3.6)	25.8 (4.2)	25.5 (4.2)
Missing	5 (5.2%)	6 (6.1%)	6 (6.1%)	17 (5.8%)
Location of primary tumor ^a^				
Right	33 (34.0%)	28 (28.6%)	22 (22.4%)	83 (28.3%)
Left	26 (26.8%)	28 (28.6%)	36 (36.7%)	90 (30.7%)
Rectum	36 (37.1%)	41 (41.8%)	38 (38.8%)	115 (39.2%)
Other	2 (2.1%)	1 (1.0%)	2 (2.0%)	5 (1.7%)
Metastasis pattern				
Synchronous	73 (75.3%)	74 (75.5%)	79 (80.6%)	226 (77.1%)
Metachronous	24 (24.7%)	24 (24.5%)	19 (19.4%)	67 (22.9%)
Metastatic sites at diagnosis				
1	54 (55.7%)	67 (68.4%)	58 (59.2%)	179 (61.1%)
>1	43 (44.3%)	31 (31.6%)	40 (40.8%)	114 (38.9%)
Liver only metastasis at diagnosis				
No	65 (67.0%)	50 (51.0%)	57 (58.2%)	172 (58.7%)
Yes	32 (33.0%)	48 (49.0%)	41 (41.8%)	121 (41.3%)
Surgery of primary tumor				
No	39 (40.2%)	24 (24.5%)	32 (32.7%)	95 (32.4%)
Yes, before metastasis	22 (22.7%)	21 (21.4%)	17 (17.3%)	60 (20.5%)
Yes, after metastasis	36 (37.1%)	53 (54.1%)	49 (50.0%)	138 (47.1%)
Metastasectomy, (any)				
No	63 (64.9%)	50 (51.0%)	45 (45.9%)	158 (53.9%)
Yes	34 (35.1%)	48 (49.0%)	53 (54.1%)	135 (46.1%)
Systemic therapy (after 1st metastasis)				
None	27 (27.8%)	29 (29.6%)	25 (25.5%)	81 (27.6%)
Fluoropyrimidines ^b^	68 (70.1%)	68 (69.4%)	72 (73.5%)	209 (71.3%)
Oxaliplatin	53 (54.6%)	60 (61.2%)	59 (60.2%)	172 (58.7%)
Irinotecan	27 (27.8%)	23 (23.5%)	21 (21.4%)	71 (24.2%)
Other chemotherapy ^c^	6 (6.2%)	4 (4.1%)	2 (2.0%)	12 (4.1%)
Bevacizumab	50 (51.5%)	52 (53.1%)	52 (53.1%)	154 (52.6%)
EGFR inhibitors ^d^	4 (4.1%)	7 (7.1%)	5 (5.1%)	16 (5.5%)
Other targeted therapy ^e^	1 (1.0%)	2 (2.0%)	1 (1.0%)	4 (1.4%)
Total days on treatment, median (IQR)	169 (83, 324)	223 (82, 370)	215 (118, 471)	211 (113, 416)

Low PA, moderate PA, and high PA refer to tertiles of weekly MET-hours, with low PA being the lowest tertile and high PA the highest tertile. Abbreviations: PA, physical activity; MET, metabolic equivalent task; IQR, interquartile range; *n*, number of patients; p2.5, 2.5th percentile; p97.5, 97.5th percentile; MVPA, moderate and vigorous physical activity; MVPA-SL, sport and leisure time moderate and vigorous physical activity; ACSM, American College of Sports Medicine, SD, standard deviation; BMI, body mass index; and EGFR, epidermal growth factor receptor. ^a^ Right: cecum, appendix, ascending colon, hepatic flexure, transverse colon; Left: splenic flexure, descending colon, sigmoid colon; Rectum: rectosigmoid, rectum; Other: not otherwise specified, overlapping. ^b^ 5-Fluorouracil, capecitabine, Tegafur/gimeracil/oteracil. ^c^ Gemcitabine/carboplatin, Trifluridine/tipiracil. ^d^ Cetuximab, Panitumumab. ^e^ Ipilimumab, Nivolumab, Pembrolizumab.

**Table 2 cancers-14-01001-t002:** Hazard ratios for overall survival according to continuous data and tertiles of physical activity categories and according to ACSM PA guideline adherence.

Determinant (Median Hours/Week)	Events/ Total	Univariate Model		Adjusted Model ^a^		Fully Adjusted Model ^b^	
		HR (95% CI)	*p* Value	HR (95% CI)	*p* Value	HR (95% CI)	*p* Value
MET							
Continuous	106/293	0.994 (0.990–0.997)	<0.001	0.995 (0.991–0.998)	0.001	0.995 (0.991–0.998)	<0.001
Low (33.1)	50/97	Ref.	Ref.	Ref.	Ref.	Ref.	Ref.
Moderate (89.9)	25/98	0.388 (0.240–0.628)	<0.001	0.448 (0.271–0.741)	0.002	0.491 (0.299–0.807)	0.005
High (166)	31/98	0.424 (0.271–0.666)	<0.001	0.491 (0.306–0.790)	0.003	0.485 (0.303–0.778)	0.003
MVPA							
Continuous	106/293	0.976 (0.958–0.994)	0.010	0.975 (0.957–0.994)	0.010	0.973 (0.955–0.992)	0.006
Low (0.5)	38/95	Ref.	Ref.	Ref.	Ref.	Ref.	Ref.
Moderate (21.6)	42/100	0.960 (0.617–1.491)	0.855	0.916 (0.575–1.459)	0.711	0.889 (0.556–1.423)	0.625
High (31.4)	26/98	0.506 (0.307–0.834)	0.008	0.491 (0.288–0.836)	0.009	0.476 (0.278–0.816)	0.007
MVPA-SL							
Continuous	106/293	0.965 (0.938–0.993)	0.015	0.955 (0.926–0.986)	0.004	0.957 (0.927–0.988)	0.007
Low (0.0)	37/90	Ref.	Ref.	Ref.	Ref.	Ref.	Ref.
Moderate (4.2)	45/105	0.909 (0.588–1.404)	0.667	0.769 (0.480–1.230)	0.273	0.737 (0.462–1.175)	0.200
High (14.0)	24/98	0.446 (0.267–0.746)	0.002	0.384 (0.223–0.661)	<0.001	0.389 (0.224–0.677)	<0.001
ACSM PA Guideline							
Non-adherence	72/181	Ref.	Ref.	Ref.	Ref.	Ref.	Ref.
Adherence	34/112	0.628 (0.417–0.945)	0.026	0.666 (0.439–1.009)	0.055	0.629 (0.412–0.961)	0.032

Abbreviations: HR, hazard ratio; CI confidence interval; MET, Metabolic equivalent task; MVPA, moderate and vigorous physical activity; MVPA-SL sport and leisure time moderate and vigorous activity; ACSM, American College of Sports and Medicine; and PA, physical activity. ^a^ Cox proportional hazard model, adjusted for baseline characteristics: age (continuous), sex (female, male), BMI (continuous), primary tumor location (left, right, rectum, other), metastatic sites (1, >1), liver-only metastasis (yes or no), synchronicity of metastasis (yes or no) ^b^ Cox model, adjusted for baseline characteristics in model 1, and additional adjustment for treatment-related factors, including surgery on primary tumor (no, yes (before diagnosis of 1st metastasis), or yes (after diagnosis of 1st metastasis)) and metastasectomy (yes or no). Seventeen participants had missing data for body mass index and were excluded from adjusted analysis.

## Data Availability

The data that support the findings of our study originate from the Prospective Dutch Colorectal Cancer (PLCRC) cohort, and clinical data were provided by the NCR. Restrictions apply to the availability of these data, which were used under license for our study. Data can be made available upon request. Access to cohort resources for future collaborative research projects may be requested through the Scientific Committee [https://plcrc.nl/for-international-visitors (accessed on: 12 January 2022)] that reviews all research projects for approval.

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
