# Peer review of "Physical Activity Is Associated with Improved Overall Survival among Patients with Metastatic Colorectal Cancer"

_cancers, 2022, doi:10.3390/cancers14041001_

Round 1
Reviewer 1 Report
Overall, this is an excellent study investigating the potential role of exercise/physical activity in an understudied cancer patient population - metastatic colorectal cancer survivors.
The only concern I had was the single measure of physical activity, which was interpreted as to improve survival. Hence, there were quite a few wording that infers causality, especially the paragraph in the discussion "Currently, no specific..." Because there was no repeated measure done on physical activity, the lower level of physical activity may as well be an indicator for a worse disease and physical condition of these patients.
If the patients were too weak to be active, they probably would have a lower tolerance of treatment, therefore the shorter time on treatment. Hence, the treatment itself might be the main drive of worse survival? Perhaps the author may consider rephrasing the messaging of their findings in the context of an observational study that included a single time measure for physical activity and avoids any language that may infer causality.
Certain the present study provides important findings to inform future studies to address the causality.
Author Response
Authors’ reply: We would like to thank the reviewer for his/her time and positive comments regarding our manuscript.
We agree that causality should not be inferred based on this single measure observational study. We checked our discussion and removed inference to causality. Also, we now mention reverse causality as our primary limitation.
We have made the following changes:
We have removed the paragraph you mentioned “Currently, no specific …”, and instead added a sentence to a later section where we also address the need for analyses with repeated measures of PA, leading to the following paragraph:
“Future analyses using repeated measures of PA and more detailed analysis of (changes in) systemic treatment will provide additional and valuable insight into associations with survival time when maintaining or increasing PA levels after mCRC diagnosis. These analyses will also inform future randomized controlled trials investigating the effects of physical activity on mCRC outcomes.
We reordered and rephrased the limitation paragraph, more explicitly stating the risk of reverse causation:
“This study has some notable limitations to consider as well. First, since this is an observational study with a single measure of PA, the association between PA and OS is at risk for reverse causation. We cannot rule out that a low level of PA is an indicator of worse disease, although we adjusted the analyses for prognostic factors. Also, analyses with exclusion of participants who died within six months did not alter our results. Still, residual confounding cannot be ruled out.”
Reviewer 2 Report
This is an interesting manuscript with interesting results, but sometimes the conclusions drawn are a bit forced, as there is no statistical evidence to support them, but only a visual inspection of the sample data. It is necessary that the authors present some more results concerning statistical inference in order to validate the conclusions that they want to generalize. Thus, I will list the points that need revision or further detail in order to improve the article for acceptance in this top quartile journal:
METHODS
Section 2.1 - Lines 67 to 77
- Authors describe the Study sample, not the Study population; please consider reformulation of the section title or of its content
- Where there any inclusion/exclusion criteria for patients? Although the study design is detailed described in reference 19, there are some basic definitions that should appear in this section (even to have the same of level of detail in all the section of this chapter - namely on sections 2.2, 2.3 and 2.4 – well done!)
Section 2.4 - Lines 111 to 128
- Please refer the level of significance used – I assume that it was 0.050, but it should be clearly stated in the manuscript
- Congratulations for the validation of the proportional hazard assumption and the use of the Schoenfeld test, it is not very common to appear although it should be performed
- I assume that authors have performed the Cox regression models in SPSS, and that they have used R and R studio to obtain the survival plots applying the survminer package, These could be clarified, in spite of not being absolutely essential, but the package used in R and the citation to it is necessary. I also apply packages available on R (and also this specific one), they are freely available to use but credits must be given to their developers.
RESULTS
Section 3.1 – Line 137
- What do you mean by notable differences? Are they statistically significant differences? Biologically relevant differences?
- Table 1 is well detailed but needs a further column to assess the p-values for comparisons between the 3 groups and identification of statistically significant differences between pairs of groups (adjusted for multiple comparisons) whenever there statistically significant difference between the three groups. The statistical tests used must be previously described in section 2.4.
- For instance, it seems that follow-up time was different in low, moderate and high PA groups, specially between the first and others. Is there any reason for this? (does mortality explain these?) Or have you used this variable as a confounder?
- Figure 1:
- I assume that the p-values presented were obtained using the log-rank test, which was not referred in the methods section 2.4
- Pairwise (adjusted) comparisons must be presented here or in an extra table otherwise authors can not state what is written is sections 3.2.1 to 3.2.4 and in discussion. These comparisons are easily obtained using the survival analysis menu in SPSS.
- Lines 158 and 159 – please refer the p-values with 3 decimal places, otherwise values between 0.0450 and 0.0549 will be undistinguished and equal to 0.05, which is assumed to be the threshold of no evidence for statistically significant differences detected.
- Table 2 – please consider using 3 extra columns to resume and present information concerning the p-values for comparing each category with the reference one in each one of the 3 models considered, and an extra one to write down the same values that you present in the survival plots presented on figure 1. The fact that there is or there is not an overlap of the 95% confidence intervals between groups is not either a necessary nor a sufficient condition for the existence of statistically significant differences between them in these set of statistical tests.
Sections 3.2.1 to 3.2.4 - Lines 177 to 205
- Need p-values for pairwise multiple comparisons between groups otherwise what is stated by the authors is not supported by the results presented; these has also impact on the discussion section (lines 213 to 253)
DISCUSSION
Lines 213 to 253
- Discussion of the results is not supported by results presented in the previous section; please perform the pairwise comparisons and present them in the previous sections, according to what was stated before
APPENDIX A
Table A
- Use the same considerations stated before for table 2 – present the p-values
Abstract
- Please review the abstract according to the previous considerations
Author Response
This is an interesting manuscript with interesting results, but sometimes the conclusions drawn are a bit forced, as there is no statistical evidence to support them, but only a visual inspection of the sample data. It is necessary that the authors present some more results concerning statistical inference in order to validate the conclusions that they want to generalize. Thus, I will list the points that need revision or further detail in order to improve the article for acceptance in this top quartile journal:
Authors’ reply: We would like to thank the reviewer for his/her time and positive comments regarding our manuscript.
METHODS
Section 2.1 - Lines 67 to 77 (now 164-182)
- Authors describe the Study sample, not the Study population; please consider reformulation of the section title or of its content
Authors’ reply: We changed the section title to ‘Study sample’. See also our response to your following remark.
- Where there any inclusion/exclusion criteria for patients? Although the study design is detailed described in reference 19, there are some basic definitions that should appear in this section (even to have the same of level of detail in all the section of this chapter - namely on sections 2.2, 2.3 and 2.4 – well done!)
Authors’ reply: All mCRC patients in the PLCRC cohort that completed a SQUASH questionnaire within 60 days of diagnosis were included in our analysis. No other inclusion criteria were used. For clarity, we have rephrased the following sentence, lines 172-173: “For the current analysis, all patients who completed a questionnaire within sixty days of diagnosis of first metastasis were included.”
Furthermore, we added a sentence about the procedure regarding diagnosis of first metastasis, lines 175-176. “Date of first metastasis was defined as date of histological confirmation or date of first imaging of metastasis if no histological proof was obtained.”
Section 2.5 - Lines 111 to 128 (now lines 218 – 238)
- Please refer the level of significance used – I assume that it was 0.050, but it should be clearly stated in the manuscript
Authors’ reply: We added the following sentence, line 206: “All statistical tests were two-sided with an alpha level of 0.050.”
- Congratulations for the validation of the proportional hazard assumption and the use of the Schoenfeld test, it is not very common to appear although it should be performed. I assume that authors have performed the Cox regression models in SPSS, and that they have used R and R studio to obtain the survival plots applying the survminerpackage, These could be clarified, in spite of not being absolutely essential, but the package used in R and the citation to it is necessary. I also apply packages available on R (and also this specific one), they are freely available to use but credits must be given to their developers.
Authors’ reply: We appreciate these comments. We used the R survival package for the Cox regression models and indeed used the survminer package for the survival plots. We have clarified this and added citations to the manuscript, lines 236-238: “Survival analyses were performed with the R survival package, version 3.2-13 [27] and survival plots were created with the R survminer package, version 0.4.9 [28]”
RESULTS
Section 3.1 – Line 137 (now removed, new paragraph starting from line 247)
- What do you mean by notable differences? Are they statistically significant differences? Biologically relevant differences?
Authors’ reply: We have added an extra table to the Appendix, showing standardized differences both between our study population and the entire mCRC population of PLCRC (which includes participants that did not give consent for receiving questionnaires) and between our study population and the general stage IV CRC population in The Netherlands.
Explanation of standardized differences is added to the methods section 2.1, lines 176-182:
“Standardized differences were calculated to quantify the magnitude of differences in patient characteristics between our study population and the general Dutch population of mCRC patients, and between our study population and all PLCRC participants with mCRC (i.e. including patients that do not consent to receiving questionnaires). Values greater than 0.20 indicate a large imbalance, while values between 0.10 and 0.20 indicate a small imbalance, and standardized differences less than 0.10 indicate a negligible imbalance [20].”
We present these results in the new Appendix A (line 641) and changed the text in section 3.1 to the following (lines 247-254): “Appendix A shows standardized differences between our study population and both the general Dutch mCRC population as well as all PLCRC participants with mCRC. Compared to the general population, our study population showed a large difference in age (63 vs 68 years), sex (67% vs 57% male) and primary tumor localization (39% vs 28% rectum). Compared to the entire PLCRC mCRC population, our study population showed a small difference in sex (67% vs 61% male) and negligible differences in age (63 vs 62 years) and primary tumor localization (39% vs 39% rectum).”
- Table 1 is well detailed but needs a further column to assess the p-values for comparisons between the 3 groups and identification of statistically significant differences between pairs of groups (adjusted for multiple comparisons) whenever there statistically significant difference between the three groups. The statistical tests used must be previously described in section 2.4.
Authors’ reply: We acknowledge the reviewer’s comment, but we would prefer to to not directly compare the three tertiles on statistical differences to comply with the Strenghtening the Reporting of Observational Studies in Epidemiology (STROBE) recommendations. Here we refer to STROBE fuideline, section 14 on descriptive data: “Inferential measures such as standard errors and confidence intervals should not be used to describe the variability of characteristics, and significance tests should be avoided in descriptive tables.”
Moreover, all variables, with the exception of systemic therapy, were used as covariates in the Cox regression models, thereby addressing the comment whether differences in characteristics have an impact on the calculated Hazard Ratios.
- For instance, it seems that follow-up time was different in low, moderate and high PA groups, specially between the first and others. Is there any reason for this? (does mortality explain these?) Or have you used this variable as a confounder?
Follow-up time is presented in Table 1 to show the average follow-up time of the entire study population and we had added it to the Cox PH models. Differences in follow-up time are indeed explained by mortality.
- Figure 1:
I assume that the p-values presented were obtained using the log-rank test, which was not referred in the methods section 2.4
Authors’ reply: These p values were indeed obtained using the log-rank test. However, our aim with Figure 1 was, besides our use of the Schoenfeld test, to serve as a visual aid to assess the proportional hazard assumption, besides our use of the Schoenfeld test. After reviewing the figure we concluded that the p values are not very informative, because it does not illustrate which groups differ significantly in case of the three PA variables that use tertiles. To avoid confusion with the reader, we decided to remove p values from the figure altogether (Line 279).
- Pairwise (adjusted) comparisons must be presented here or in an extra table otherwise authors can not state what is written is sections 3.2.1 to 3.2.4 and in discussion. These comparisons are easily obtained using the survival analysis menu in SPSS.
Authors’ reply: The results in section 3.2.1 to 3.2.4 are based on the Cox proportional hazard regression analysis. In regards to the three physical activity variables for which we created tertiles, we independently compare the two higher tertiles (moderate and high) with the reference tertile (Low).
However, we agree that we do not directly answer whether an increase in physical activity is associated with improved OS. We therefore performed additional analyses, for which we used hours per week as continuous variables for MET-hours, MVPA and MVPA-SL and we found that every hour increase in PA was associated with improved OS. We updated the methods with the following sentence (lines 220-223): “PA was assessed using continuous data and tertiles for weekly MET-hours, MVPA and MVPA-SL (henceforth: tertile 1 = low; tertile 2 = moderate; tertile 3 = high level of PA), with low being the reference category.”
And we updated the results, by expanding table 2 (lines 294-295): “Table 2. Hazard ratios for overall survival according to continuous data and tertiles of physical activity categories and according to ACSM PA guideline adherence.”
We also updated the according sections (3.2.1 to 3.2.3, lines 304-342) with the following:
MET-hours (Lines 305-308): “An increase in weekly MET-hours was consistently significantly associated with improved survival across all models (univariate HR 0.994 [95% CI 0.990-0.997, p value<0.001]; adjusted HR 0.995 [95% CI 0.991-0.998, p value=0.001]; fully adjusted HR 0.995 [95% CI 0.991-0.998, p value<0.001]).”
MVPA (Lines 320-322): “An increase in weekly hours spent on MVPA was consistently significantly associated with improved survival across all models (univariate HR 0.976 [95% CI 0.958-0.994, p value=0.010]; adjusted HR 0.975 [95% CI 0.957-0.994, p value=0.010]; fully adjusted HR 0.973 [95% CI 0.955-0.992, p value=0.006]).”
MVPA-SL (Lines 332-334): “An increase in weekly hours spent on MVPA-SL was consistently significantly associated with improved survival across all models (univariate HR 0.965 [95% CI 0.938-0.993, p value=0.015]; adjusted HR 0.955 [95% CI 0.926-0.986, p value=0.004]; fully adjusted HR 0.957 [95% CI 0.927-0.988, p value=0.007]).”
- Lines 158 and 159 (now 277-279) – please refer the p-values with 3 decimal places, otherwise values between 0.0450 and 0.0549 will be undistinguished and equal to 0.05, which is assumed to be the threshold of no evidence for statistically significant differences detected.
Authors’ reply: We have added an extra decimal, lines 277-279 : “Based on the Schoenfeld test, proportionality assumption was met for MET-hours (p value=0.202), MVPA (p value=0.164), MVPA-SL (p value=0.294), and guideline adherence (p value=0.257).”
- Table 2 – please consider using 3 extra columns to resume and present information concerning the p-values for comparing each category with the reference one in each one of the 3 models considered, and an extra one to write down the same values that you present in the survival plots presented on figure 1. The fact that there is or there is not an overlap of the 95% confidence intervals between groups is not either a necessary nor a sufficient condition for the existence of statistically significant differences between them in these set of statistical tests.
Authors’ reply: We have added the 3 extra columns with p values to table 2 for the univariate, adjusted and fully adjusted models.
In accordance with our comment regarding removal of the p values in the survival plots, we prefer to not present them here either.
Sections 3.2.1 to 3.2.4 - Lines 177 to 205 (Now Lines 271 to 311
- Need p-values for pairwise multiple comparisons between groups otherwise what is stated by the authors is not supported by the results presented; these has also impact on the discussion section (lines 213 to 253)
Authors’ reply: We added p values to the results section (3.2.1 – 3.2.4, lines 304 – 347) and as stated above we added the analyses using PA as a continuous variable to investigate whether every increase in PA is associated with a decrease in risk of shorter survival.
DISCUSSION
Lines 213 to 253 (Now starting from line 360)
- Discussion of the results is not supported by results presented in the previous section; please perform the pairwise comparisons and present them in the previous sections, according to what was stated before
Authors’ reply: We kindly refer to our answer to previous comments. The additional continuous analyses of PA support our findings. We changed the first paragraph to include those findings (Lines 360-367):
“In this prospective observational cohort of mCRC patients, we found that higher weekly total PA (MET-hours), MVPA and MVPA-SL at diagnosis of first metastasis was significantly associated with prolonged survival time compared to low levels. When comparing tertiles, significant associations were seen for high levels of MVPA and MVPA-SL, and high and moderate levels of total PA (MET-hours per week) Significant associations with increased survival time were also seen for ACSM PA guideline adherence compared to non-adherence. Greatest risk reductions were seen for high levels of MVPA-SL compared to low levels.”
APPENDIX A (Now Appendix B, line 667)
Table A
- Use the same considerations stated before for table 2 – present the p-values
Authors’ reply: We have added the 3 extra columns with p values to the Appendix (now Appendix B) and added the continouous data as rows, in accordance with our changes to table 2.
Abstract
- Please review the abstract according to the previous considerations
Authors’ reply: We have reviewed the abstract and made minor changes: we specified that the presented tertile results are in comparison with the lowest tertile and we added p values (with the correct amount of decimals). Due to the limited word count, we decided not to include the findings for the continuous PA determinants
Round 2
Reviewer 2 Report
The authors provided good answers and the manuscript has been sufficiently improved